# Experimental demonstration of quantum advantage for NP verification with limited information

Federico Centrone [1,2 ✉], Niraj Kumar[3 ✉], Eleni Diamanti [1 ✉] & Iordanis Kerenidis[2,4,5 ✉]

In recent years, many computational tasks have been proposed as candidates for showing a quantum computational advantage, that is an advantage in the time needed to perform the task using a quantum instead of a classical machine. Nevertheless, practical demonstrations of such an advantage remain particularly challenging because of the difficulty in bringing together all necessary theoretical and experimental ingredients. Here, we show an experimental demonstration of a quantum computational advantage in a prover-verifier interactive setting, where the computational task consists in the verification of an NP-complete problem by a verifier who only gets limited information about the proof sent by an untrusted prover in the form of a series of unentangled quantum states. We provide a simple linear optical implementation that can perform this verification task efficiently (within a few seconds), while we also provide strong evidence that, fixing the size of the proof, a classical computer would take much longer time (assuming only that it takes exponential time to solve an NP-complete problem). While our computational advantage concerns a specific task in a scenario of mostly theoretical interest, it brings us a step closer to potential useful applications, such as server-client quantum computing.

---

[1] Sorbonne Université, CNRS, LIP6, Paris, France. [2] Université de Paris, CNRS, IRIF, Paris, France. [3] School of Informatics, University of Edinburgh, Edinburgh, UK. [4] QC Ware Corp, Palo Alto, CA, USA. [5] QC Ware Corp, Paris, France. ✉email: federico.centrone@lip6.fr; nkumar@exseed.ed.ac.uk; eleni.diamanti@lip6.fr; jkeren@irif.fr

Quantum technologies explore the possibility of using quantum resources in order to demonstrate in practice an advantage in terms of computational time, security or communication efficiency. A series of proposals of tasks for which a computational advantage can be shown have appeared, including Boson Sampling[1,2], which has been implemented for small sizes[3–7], and sparse commuting (IQP) or random quantum circuits[8–14]. The quest for such a quantum computational advantage has culminated recently with a demonstration of a random circuit sampling task by Google using the 53-qubit superconducting chip Sycamore[15].

One of the major difficulties in gaining confidence with these experimental demonstrations, and cause for some doubts (see Ref. [16]), is that these are not well established tasks for which classical methods have been developed for long, thus making benchmarking against classical methods difficult. In particular, although the asymptotic theoretical separation between quantum and classical methods is based on strong computational complexity theoretic assumptions (namely, polynomial hierarchy collapses at the third level), this is less clear when considering the exact scaling of the optimal classical algorithm to solve the task at intermediate sizes and in the presence of noise. Moreover, the verification of the advantage provided by the quantum machine can only happen for a very narrow range of parameters where the classical complexity is just out of reach but some kind of verification (usually of smaller or simpler instances) is still possible to perform on a classical computer. Last, the main open question is to demonstrate such superior behavior for a useful task, thus proving the disruptive potential of quantum technologies.

In this work, we study the power of quantum technologies to provide a computational advantage in an interactive setting, where first we allow two parties to interact in a predefined manner, and then we look at the time it takes for one of them to resolve a specific computational task when they can use quantum or classical resources. Specifically, we study the task of verifying NP-complete problems, in particular whether a set of boolean constraints have a satisfying assignment to them or not, when an untrusted party provides some limited information about the solution of the problem. For this task, we show that we can achieve a quantum advantage exploiting experimental techniques involving coherent states, linear optics and single-photon detection.

Before explaining this further let us remark a few properties of our result: first, the quantum hardware we use is simple and the demonstration can be readily reproduced in well-equipped quantum photonics labs; second, our task is inherently verifiable since the output is a YES/NO answer and not a sample from an exponential size distribution (we emphasize here that the quantum machine in our scenario is certainly not solving NP-complete problems but merely verifies whether a solution exists or not with limited information about the possible solution); third, the benchmarking against the best classical methods is based only on the assumption that NP-complete problems do not have sub-exponential algorithms, a well-known and widely accepted computational assumption[17]; and finally, while previously experimentally demonstrated computational tasks are typically tailor-made for showing quantum advantage with no direct connection to useful applications, the fast verification of NP-complete problems with bounded information leakage could potentially lead to interesting applications, including in server–client quantum computing, authentication systems, ethical behavior enforcement, and blockchain technologies[18]. At the same time, we stress that the computational advantage we achieve is not in the standard computational model where a single classical or quantum machine receives an input and computes an output, but in the interactive setting, where we first allow interaction with a second party before trying to resolve the computational task at hand.

Let us now describe our results on the demonstration of a quantum computational advantage in this interactive setting for NP verification in more detail.

The class of NP-complete problems contains some of the most interesting problems both from a theoretical point of view and in practice. Such problems include the Traveling Salesman Problem, Satisfiability, and many problems related to combinatorial optimization, scheduling, networks, etc. The main characteristic of these problems is that while it is very difficult to find a solution, and in many cases even approximate the optimal solution, it is easy to verify a solution if someone provides one to us, even if this is an untrusted party. Moreover, the theory of NP-completeness shows that all these different problems are related to each other through reductions, meaning that it suffices to study one of them in order to say something interesting about the entire class of problems.

Let us then focus on 2-out-of-4 SAT, which can be obtained through a reduction of a 3-SAT, the canonical NP-complete problem. The 2-out-of-4 SAT problem consists of a formula of $N$ boolean variables in a conjunction of clauses, where each clause is satisfied if and only if exactly two of the four variables forming the clause are True. The task is to decide whether there exists an assignment to the variables $(x_1, x_2, \ldots, x_N)$, which satisfies all clauses of the formula, in other words for every clause two variables must be True and the other two must be False. We assume without loss of generality that our 2-out-of-4 SAT instance meets the following two conditions. First, it is a balanced formula, meaning that every variable occurs in the same constant number of clauses, and second, it is a Probabilistically Checkable Proof (PCP), i.e., either the formula is satisfiable, or for any assignment at least $\delta$ fraction of the clauses is unsatisfiable, for some constant $\delta > 0$. These conditions can always be guaranteed using a polynomial overhead in $N$ and the theory of PCPs. Thus any NP-complete problem can be reduced to a balanced 2-out-of-4 SAT instance that is probabilistically checkable.

For the verification of such a 2-out-of-4 SAT instance, we would like the verifier, Arthur, to accept a correct proof (a truth assignment of the variables that satisfies the formula) given by a prover, Merlin, with high probability, say $\mathcal{C} \geq 2/3$. We call this the completeness property of the verification scheme. If, on the other hand, the formula is not satisfiable, then for any potential proof he receives, Arthur must accept the proof with low probability, say $\mathcal{S} \leq 1/3$. This is the soundness property of the verification scheme. For a 2-out-of-4 SAT problem of size $N$, the best algorithms for finding a solution run in time exponential in $N$ (using some sort of clever brute force search for a solution)[19], while the verification of a potential solution takes time linear in $N$. One important property of NP-complete problems is that if we accept that the best algorithms for solving an NP-complete problem are exponential in $N$, then if one has found or has been provided with part of a solution, for example the truth assignment to a subset of the variables of size $t < N$, then in the worst case the remaining time to complete the solution is still exponential in $(N − t)$[17].

The use of quantum protocols for verification in this so-called interactive proof setting was first employed in Ref. [20], which introduced the concept of Quantum Merlin Arthur. Since then, QMA problems have been intensively studied[21–25]. They are the quantum analog to NP problems in computational complexity theory and have the same completeness and soundness properties as the ones described above with the proofs encoded in quantum states.

By the results of Ref. [25], we know that quantum Merlin Arthur interactive proof systems can be used to verify NP-complete problems more efficiently than the classical ones. In particular, it

was shown that a quantum verifier who receives $O(\sqrt{N})$ unentangled copies of a quantum proof can verify efficiently the 2-out-of-4 SAT instance by performing a number of tests/measurements on these states. Note that the assumption that the proofs are unentangled is crucial. Here, the quantum proof is the state $\frac{1}{\sqrt{N}}\sum_{i=1}^{N}(-1)^{x_i}|i\rangle$, i.e., the quantum state on $\log_2 N$ qubits encoding the values of the assignment $(x_1, \ldots, x_N)$ as amplitudes. The information Arthur receives about the classical solution cannot be more than $O(\sqrt{N}\log_2 N)$ bits of information, since this is the number of qubits he receives, nevertheless, the verification becomes efficient in the quantum case: for the same amount of revealed information a classical verification protocol would require exponential time while it takes polynomial time for the quantum protocol to perform the task. We remark that one can see the quantum advantage either as a computational advantage, as we do in our work, where we ask how long the verification will take in the quantum and classical case if we fix the size of the message sent by the prover, or as an information advantage, where we ask what size of quantum or classical message is needed if we fix the time of the verification to be polynomial in the input size. We stress again that, in both cases, the advantage is not about solving NP-complete problems, but about verifying them with limited information. In Ref. [26], it was first shown that in theory it is possible to implement such a verification protocol with single photons and linear optics, albeit a practical implementation is and will probably continue to be out of reach for photonics technology due to the extremely large number of elements in the proposed scheme.

Here, we overcome this limitation by proposing a quantum verification test that maintains the properties of the original one and at the same time uses new conceptual tools that make it practical. This allows us to provide the first experimental demonstration of an efficient quantum verification scheme for NP-complete problems, and hence a strong provable quantum advantage for this task based on the assumption that finding a solution to NP-complete problems takes exponential time on a classical computer. More precisely, we experimentally demonstrate how a quantum Arthur who receives an unentangled quantum proof of size $\widetilde{O}(N^{3/4})$ (where $\widetilde{O}$ denotes the order up to logarithmic terms) can verify 2-out-of-4 SAT instances in time linear in $N$, while a well-known assumption is that any known classical algorithm takes time exponential in $(N - \widetilde{O}(N^{3/4}))$. The core idea of our protocol that enables us to perform the verification with coherent states and a simple linear optics scheme is based on the Sampling Matching problem defined and implemented in Ref. [27]. This is particularly appealing from a practical point of view because of the relative ease of preparation and manipulation of coherent states, which combined with linear optics transformations have made them attractive candidates for proving quantum advantage in communication complexity and security [28–34]. The use of the Sampling Matching is also one of the main conceptual differences of our current protocol with respect to the work of Ref. [26], which provided a verification protocol with single photons and which cannot readily be made to work simply by mapping the single photons into coherent states.

In order to explain the importance of our result let us first go back to the classical case and describe a possible scheme for verification. Since we know that in case the formula is not satisfiable then for any assignment at least a constant $\delta$ fraction of the clauses are not satisfied, then for verification it suffices for Arthur to pick a random clause, obtain the values of the four variables and check whether the clause is satisfied or not. By repeating this for a small constant number of clauses, Arthur can verify with high probability whether the instance is satisfiable or not, and moreover, the information Arthur receives about the

solution is very small (just the value of the variables in a few clauses). We can also see this protocol in a slightly modified version, which will be closer to our quantum verification protocol based on Sampling Matching. Instead of having Arthur pick uniformly at random a small number of clauses out of all possible clauses to verify, we can assume that Arthur picks each clause with some probability so in the end the expected number of clauses he picks is the same number as in the initial protocol.

There is of course a well-known issue in these schemes. Once Merlin knows which clause Arthur wants to test, he can easily adapt the values of the variables to make this clause satisfiable. Arthur cannot force Merlin to be consistent across the different clauses, namely to keep the same value for each variable in the different clauses. One way to remedy this would be by having Merlin send the entire assignment to Arthur (which is the usual verification protocol), but in this case Arthur gets all the information about the classical solution. Another solution is through interactive computational zero-knowledge proofs, where one uses cryptographic primitives, i.e., bit commitment, in order to force the behavior of Merlin, but such schemes necessitate communication between Arthur and Merlin and only offer computational security [35].

Thus in the classical world, it is impossible to have a protocol with a single message from Merlin to Arthur that performs verification while at the same time Arthur does not learn the entire classical solution.

In the quantum world, using coherent states and a new efficient linear optics scheme based on the Sampling Matching, we can experimentally demonstrate exactly that: a quantum Arthur can efficiently verify instances of NP-complete problems (in time linear in the size $N$) while at the same time receiving only a small amount of information about the solution (theoretically of order $\widetilde{O}(N^{3/4})$). To show this advantage experimentally it was sufficient to use sequences of a few thousand coherent pulses, corresponding to a proof size $N$ from 5000 to 15,000, with an average mean photon number per pulse on the order of 1, and standard InGaAs single-photon detectors.

We are now ready to give the details of our quantum verification protocol, analyze its completeness and soundness, and provide the results of our experimental demonstration.

## Results

**Quantum proofs encoded in coherent states.** In the first step of our verification protocol, Merlin sends the quantum proof to Arthur. We consider here that if the instance is satisfiable then an honest Merlin will use coherent states to encode the proof, exploiting the coherent state mapping introduced in Refs. [28,29]. More precisely, he encodes his proof $x = (x_1, x_2, \ldots, x_N)$ in a time sequence of $N$ weak coherent states. He does this by applying the displacement operator $\hat{D}_x(\alpha) = \exp(\alpha\hat{a}_x^\dagger - \alpha^*\hat{a}_x)$ to the vacuum state, where $\hat{a}_x = \frac{1}{\sqrt{N}}\sum_{k=1}^{N}(-1)^{x_k}\hat{a}_k$ is the annihilation operator of the entire coherent state mode, and $\hat{a}_k$ is the photon annihilation operator of the $k$th time mode. Hence,

$$|\alpha_x\rangle = \hat{D}_x(\alpha)|0\rangle = \bigotimes_{k=1}^{N}|(-1)^{x_k}\alpha\rangle_k, \qquad (1)$$

where $|(-1)^{x_k}\alpha\rangle_k$ is a coherent state with mean photon number $\mu = |\alpha|^2$ occupying the $k$th time mode. Thus, the state $|\alpha_x\rangle$ has a mean photon number $|\alpha_x|^2 = N|\alpha|^2$, with the photons distributed over the entire sequence of $N$ modes. Note that varying the parameter $\alpha$ controls how many photons are expected to be in the state; for example for $\alpha = 1$, every coherent state in the sequence has on average one photon, while if we take $\alpha = 1/\sqrt{N}$, then on average only one photon will be present in the entire sequence.

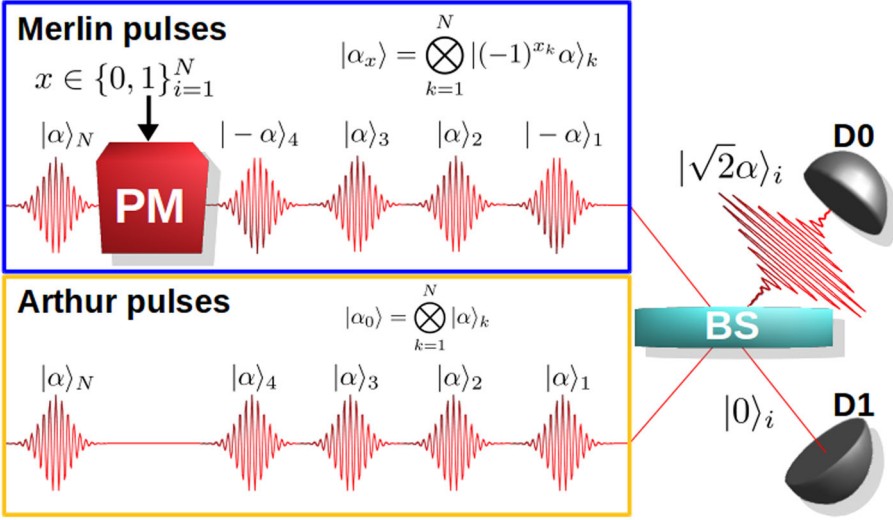

**Fig. 1 The Sampling Matching scheme (SM).** Merlin creates his coherent state quantum proof by sequentially encoding his proof $x$ into the coherent pulses. Under the SM scheme, Arthur interferes Merlin's coherent state quantum proof with his local state consisting of a sequence of $N$ pulses. He observes the clicks in two single-photon threshold detectors $D_0$ and $D_1$ to decide whether Merlin's proof state is correct.

In the single-photon version of the original protocol[25,26], Merlin prepares $O(\sqrt{N})$ unentangled copies of a state that consists of a single photon in $N$ modes, i.e., a state in an $N$-dimensional Hilbert space. This implies that during the protocol the information revealed to Arthur is at most $O(\sqrt{N}\log_2 N)$ bits of information. Then, a number of tests are performed on these states to check that they are equal, uniform, and that they satisfy the boolean formula. For the equality, a SWAP test is performed between different copies of the proofs; for testing that the amplitudes of the states are roughly uniform, a test based on the Hidden Matching problem is performed; for satisfiability, the parity of four variables that belong to the same clause is measured in order to check whether the specific clause is satisfied. Each test is performed with some probability and if the test is successful, then Arthur accepts the instance as satisfiable.

An important feature of our protocol is that by using the Sampling Matching method we are able to combine the above tests into a single test and all copies of the proofs into a higher mean photon number sequence of $N$ coherent states, which we also assume to be unentangled. By sending coherent states with a higher mean photon number $|\alpha|^2$ we essentially increase the probability of measuring each variable and thus the information conveyed by Merlin; this is important for the uniformity and satisfiability parts of our verification test as we will see later. Increasing $|\alpha|^2$ instead of sending multiple copies of the same state also allows us to avoid the necessity of applying the equality test that was ensuring that the copies are the same. On the other hand, the unentanglement assumption for the sequence of coherent pulses is necessary as it was in Refs. [24,25], since otherwise this would lead to a subexponential quantum algorithm for solving NP-complete problems, which is thought to not be possible.

We prove in the following that theoretically the average photon number for each of the $N$ coherent states that the honest Merlin sends when the instance is satisfiable is of the order of $|\alpha|^2 = O(N^{-1/4})$, which makes the information Arthur gets about the classical solution to be $\widetilde{O}(N^{3/4})$. In high level, this also implies that any classical verification algorithm with the same amount of information will take time exponential in $(N - \widetilde{O}(N^{3/4}))$, which becomes large enough for practical sizes of $N$. This is because Arthur can always enumerate over all possible proofs Merlin

sends and perform the verification for each one of them. It will take him time exponential in $\widetilde{O}(N^{3/4})$ to enumerate over all possible proofs (since the information in them is less than $\widetilde{O}(N^{3/4})$) and thus if the verification for each of them takes time less than exponential in $(N - \widetilde{O}(N^{3/4}))$ then this would imply a fast algorithm for NP.

Once Arthur receives the quantum proof as a sequence of unentangled coherent states from Merlin, he performs the verification by applying a verification test. We assume that Merlin can behave dishonestly in any way possible, apart from having to send unentangled states. Let us now describe this verification test and how it can be performed in a linear optical setting.

**Verification test.** As we discussed previously, the original verification test[25] consists in first testing that the copies of the proofs are the same (which we have avoided by sending a single sequence of coherent states), and then that the amplitudes of each of these states are close to uniform. This test is necessary in order to show that Arthur can actually check all possible clauses with roughly uniform probability. Otherwise, Merlin can just force Arthur to always measure some specific subset of variables (the ones that can satisfy some corresponding subset of clauses) and thus convince Arthur of the validity of the assignment, even though no assignment exists that satisfies all clauses.

Here, we deal with this in a different way. Again, we want to ensure that Arthur will measure each clause with some probability, meaning that Merlin cannot force Arthur to measure only a specific subset of variables and clauses. This is where we use the idea of Sampling Matching[27], which was introduced as a practical version of Hidden Matching, the problem performed in the original uniformity test. Instead of interfering Merlin's coherent states with themselves, we in fact input in an interferometer Merlin's sequence of coherent states in one arm, and a new sequence of coherent states prepared by Arthur in the other arm. This is also the main difference with the single-photon protocol in Ref. [26].

More specifically, the test as depicted in Fig. 1 is the following. When Arthur receives the state $|\alpha_x\rangle$ from Merlin with the mean photon number $|\alpha_x|^2$ predefined by the protocol, he generates his local state in the form of a sequence of uniform coherent pulses,

with the same mean photon number. In particular, Arthur creates the state

$$|\alpha_0\rangle = \bigotimes_{k=1}^{N} |\alpha\rangle_k, \qquad (2)$$

such that $|\alpha_0|^2 = |\alpha_x|^2$. He then sequentially interferes each of honest Merlin's coherent states with his local coherent states in a balanced beam splitter and collects the outputs in the two single-photon detectors, $D_0$ and $D_1$. At each time step $k$, the input state in the beam splitter is $|(-1)^{x_k}\alpha\rangle_k \otimes |\alpha\rangle_k$, while at the output modes we have,

$$\left|\frac{((-1)^{x_k}+1)\alpha}{\sqrt{2}}\right\rangle_{D_0,k} \otimes \left|\frac{((-1)^{x_k}-1)\alpha}{\sqrt{2}}\right\rangle_{D_1,k}. \qquad (3)$$

Then, the probability of getting a click on each of the single-photon detectors at the $k$th time step of the verification protocol is:

$$P_{\text{det}}^{(k)} = \begin{cases} 1 - e^{-|\alpha|^2((-1)^{x_k}+1)^2/2} & \text{on } D_0 \\ 1 - e^{-|\alpha|^2((-1)^{x_k}-1)^2/2} & \text{on } D_1. \end{cases} \qquad (4)$$

One way of understanding the above test is to note that it is guaranteed that Arthur receives a value for each variable with at least some probability, due to the photons in his own state. This way, Merlin cannot choose exactly for what variables Arthur will obtain a value. Thus, Arthur will end up obtaining the values of a subset of variables that is random enough (meaning Merlin cannot deterministically choose it) so that when he considers the clauses whose variables are in this subset, then either all of them will be satisfied in the YES-instance, or sufficiently many of them will not be satisfied in the NO-instance.

Now, if Merlin wants to send a value for a specific variable $x_k$ to Arthur, then he can do it perfectly, since by constructing an honest coherent state of the form $|(-1)^{x_k}\alpha\rangle_k$, only one of the two detectors of Arthur has non-zero probability of clicking. On the other hand, if Merlin sends any state $|\beta\rangle$, then after the interaction with Arthur's coherent state $|\alpha\rangle$ one important thing is true: no matter what Merlin's state is, there is still a probability of a detector click, which is at least $1 - e^{-|\alpha|^2}$ due to the photons in Arthur's coherent state and the fact that we only perform linear optics operations that preserve the number of photons. In other words, Arthur obtains a value for each variable with some probability independent of Merlin's message, and this value can be fixed by Merlin if he honestly sends a state that encodes a value.

After recording the results of his measurements, Arthur assigns values to the variables in the following way: if the detector $D_0$ clicked, then the value is 0, if the detector $D_1$ clicked then the value is 1, while he leaves the variables unassigned if no click was observed. Then, Arthur checks for each clause for which he has assigned a value to all four variables whether it is satisfied or not, namely if exactly two out of the four variables in the clause have value 1. In the ideal case where there are no errors, Arthur will accept if all clauses are satisfied and reject if any clause is not satisfied. In the presence of non-ideal experimental conditions, we will see that Arthur will use a threshold and accept if at least that fraction of clauses are satisfied or else he will reject.

We are now ready to analyze the completeness of the protocol, namely the probability Arthur accepts assuming that the 2-out-of-4 SAT instance is a satisfiable instance, in which case Merlin prepares a proof state in the form $|\alpha_x\rangle$ for a satisfying assignment $x$. Then, we discuss the soundness of the protocol, namely the case in which the instance has no satisfying assignment and where Merlin still wants Arthur to accept his proof and acts

dishonestly. He will then try to send some general quantum state to trick Arthur, while, as we said, here we make the same type of assumption as in the original work of Aaronson et al.[25], namely that Merlin still sends a sequence of unentangled states. Later, we will complete the analysis by looking at the protocol under non-ideal experimental conditions and see what level of noise the interferometric setup can tolerate in order to maintain a positive gap between the completeness and soundness probabilities.

The completeness corresponds to the probability that Arthur accepts the proof of Merlin in the case of a satisfiable instance, where Merlin sends the correct quantum state. As we have described, Arthur will retrieve the values of a number of variables that are encoded in the phases of Merlin's sequence of coherent states by using his own local coherent states and the interferometric setup shown in Fig. 1. As long as Merlin honestly encodes the satisfying assignment into his coherent states then only one detector has non-zero probability of clicking and thus Arthur will never get a wrong value. Thus the only probability of rejecting comes from Arthur not obtaining the values of the four variables of any clause.

To estimate this probability, and hence the completeness, we remark again that the unentanglement promise guarantees that the probability of detecting a photon in each of the pulses in the sequence is independent of the remaining pulses of the sequence, since the pulses are unentangled between them. Furthermore, the probability of measuring a particular variable is independent of which clause Arthur is going to verify later on. If we now denote as $p_h \geq 1 - e^{-2|\alpha|^2}$ the probability that a detector clicks during a time step in an honest run (see Eq. (4)), then the probability that a specific clause is measured (meaning all four variables in the clause are measured) is at least $p_h^4$ (where we have used the independence remarks above).

We have also assumed that the instance is balanced and each variable appears in a constant number of clauses, which implies that the number of clauses in an instance of the problem is $O(N)$.

Taking into account the above, we see that the probability that Arthur does not obtain the values of the four variables for any clause in an instance is at most $(1 - p_h^4)^{O(N)}$. This can be made arbitrarily small, and therefore the completeness arbitrarily close to 1, as long as $p_h^4 = O(N^{-1})$ for a large enough constant, which in turn implies that it suffices to take $|\alpha|^2$ on the order of $O(N^{-1/4})$ with a large enough constant. Note that by taking $|\alpha|^2$ on the order of $O(N^{-1/4})$, the verifier is expected to receive $\widetilde{O}(N^{3/4})$ clicks in the detectors. This is higher than the $O(\sqrt{N})$ bits in the original protocol of Ref. [25], where one can choose a specific measurement (depending on the clauses) to always get the value of a clause and hence check satisfiability. In fact the $O(\sqrt{N})$ is needed to prove the uniformity of the state. In our case, the way we achieve a good probability of measuring all variables in a clause of the instance is by increasing the $|\alpha|^2$ to ensure we are measuring enough variables, and that number needs to be now $\widetilde{O}(N^{3/4})$ to make the probabilities work out. We will see later that experimentally we will pick specific values for $N$ and $|\alpha|^2$ that keep the completeness higher than 0.9.

We are going to show now that if the 2-out-of-4 SAT is a NO instance, then the soundness of the protocol, namely the probability of Arthur accepting the proof, is small enough no matter the strategy of the prover as long as the promise of unentanglement holds. For this, we highlight again two important features of our test and the properties of the SAT instances we are dealing with. First, at least a $\delta$ fraction of the clauses are unsatisfiable for any assignment of variables, and second, the probability of measuring a particular variable is lower bounded by the fact that Arthur inputs an honest coherent state into the

interferometer, even if Merlin sends no photon in his corresponding state.

We can then bound the probability that Arthur measures the values of some variables and finds a clause that contains them and is not satisfied. We have already seen that the minimum probability of Arthur obtaining a value for any variable, no matter what Merlin sends, is $p_d \geq 1 - e^{-|\alpha|^2}$. Then, following the same rationale as before, since a constant $\delta$ fraction of clauses are unsatisfied for any assignment, we can conclude that the probability of measuring the values of four variables that make a clause unsatisfied is at least $\delta p_d^4$. Assuming again that there are $O(N)$ clauses in an instance, the probability that Arthur does not find any unsatisfied clause is at most $(1 - \delta p_d^4)^{O(N)}$. So again we just need to pick $|\alpha|^2$ large enough in order to make the soundness small enough. In particular, since $\delta$ is a constant, we can pick as before $|\alpha|^2 = O((\delta N)^{-1/4}) = O(N^{-1/4})$ and make soundness arbitrarily small. We will see later that experimentally we will pick values for $N$ and $|\alpha|^2$ that keep the soundness lower than 0.6.

**Classical complexity of verification**. Our quantum verification test takes time $O(N)$ to implement, since Arthur receives a sequence of $N$ pulses that he interferes with his own coherent states and then he simply calculates the number of satisfied clauses (from the $O(N)$ of them) before accepting or rejecting. To compare our test with classical resources in terms of complexity, we are making here a well founded assumption that any classical algorithm for solving 2-out-of-4 SAT runs in time exponential in the instance size $N$. In particular, we consider the classical complexity to be of the form $2^{\gamma N}$ for some constant $\gamma \leq 1$. Shöning's algorithm for 3-SAT takes time $(4/3)^n$ on average for instances of size $n$[36], while the best-known practical SAT solvers can provide a complexity of $O(1.307^N) = O(2^{0.4N})$[19]. We have also discussed previously that if the information that Arthur gets about the proof is $t$ bits, then the running time of the classical algorithm remains exponential in $(N - t)$.

The value of $t$, namely the bits of information Arthur obtains about the proof during the verification of a YES instance, can be easily upper bounded for our test by the number of detector clicks during the verification procedure. We want to remark here that in our setting we have an honest Arthur who tries to verify the instance and we do not have to consider a cheating Arthur as in the case of standard cryptographic settings. The expected number of clicks in Arthur's detectors depends on the parameter $|\alpha|^2$, namely the average number of photons per pulse. In particular we have that the number of clicks is $O(N(1 - e^{-2|\alpha|^2}))$ and for our value of $|\alpha|^2 = O(N^{-1/4})$ we have that the information obtained by Arthur is at most $\widetilde{O}(N^{3/4})$. Thus, by picking large enough $N$ it is easy to make the difference $(N - \widetilde{O}(N^{3/4}))$ also large enough.

We will see later that experimentally we will keep this difference larger than 1000. This is an arbitrary choice that nonetheless is more than sufficient to confirm that the classical computation would be unfeasible. For example, given a difference of 150, we can calculate that we would need a 45-digit number of operations to verify the SAT instance: even with processors working at 10 GHz and operated by 10 billion people, and repeating the operation in 10 billion planet Earth copies, parallelizing somehow the whole process, it would be necessary to wait around the age of the Universe to be able to classically verify such instances.

To summarize the above, in the setting that we have described we define the notion of quantum advantage for verifying NP-complete problems of size $N$ with bounded information when three conditions are fulfilled:

1. The verification of the proof by a quantum Arthur takes time linear in $N$;
2. The obtained completeness is high enough and soundness low enough, where in our case we have set $\mathcal{C} > 0.9$ and $\mathcal{S} < 0.6$;
3. The number of bits of information on the proof that Arthur obtains is much smaller than $N$, in our case at least 1000 bits smaller, so that the classical complexity of performing the same task is such that it is effectively unfeasible.

**Dealing with practical imperfections**. Let us now consider how we can take into account practical imperfections in our verification test in view of its experimental implementation for demonstrating a quantum advantage as we have defined it above.

Up till now we have assumed that Arthur measures the values of the variables perfectly when Merlin is honest. In a practical setting, however, this may not be the case due to errors coming mainly from the imperfect visibility of the interferometric setup and the finite quantum efficiency and dark counts of the single-photon detectors.

There is a simple way to remedy the verification test in order to deal with such imperfections. Arthur performs the same measurements and assigns values to the variables in the following way: when only one detector clicks then he assigns the corresponding value to the variable, i.e., he assigns the value 0 if he registers a click in detector $D_0$ and nothing in $D_1$ and vice versa; when both detectors click (which can occur in practice due to the imperfections) then he assigns a uniformly random value to the variable; when no detector clicks then the variable remains unassigned. Note that the fact of picking a random value for a variable in case of double clicks, instead of ignoring this variable, helps avoiding the case where Merlin would input a large number of photons to force double clicks for the variables that he would not want Arthur to measure. Once Arthur assigns the values to the variables, he looks at the clauses for which all four variables have been assigned a value and checks if the clause is satisfied, namely if exactly two out of four variables have the value 1. Knowing the experimental parameters, we can calculate the expected fraction of satisfied clauses in the YES instance (which should be only slightly less than 1 for photonic systems with low loss and errors) and the one in the NO instance (which should be much less than 1 for instances with large enough $\delta$ and small enough errors). Arthur can now define an appropriate threshold for the number of satisfied clauses above which he accepts and below which he rejects, and assuming an appropriate gap between the number of satisfied clauses in the YES and NO instances we can then guarantee a large gap between completeness and soundness using simple Chernoff bound calculations.

We will try now to find an experimental parameter regime where we can show quantum advantage. For this, we first make one more assumption about the dishonest Merlin, which is that he always sends states that have the correct mean photon number $\mu = |\alpha|^2$ specified by the protocol, while he can freely choose the assignment values in order to trick Arthur to accept. Note that here we are not trying to define a general interactive proof (Arthur-Merlin) system; we are trying to construct a specific computational task for experimentally demonstrating quantum advantage. Thus, we add on top of the unentanglement assumption the assumption of states with the appropriate mean photon number so as to make the implementation of this task simpler. This essentially corresponds to a dishonest Merlin who can only cheat "classically", in the sense that he can choose whatever assignment he wants for the variables encoded in the quantum states and then send states of the form in Eq. (1) (see also Fig. 1). This is an assumption that is only needed in order to

perform our proof of principle experiment but it is not needed for any of the previous analysis of the protocol, including about completeness and soundness. In fact, even without this assumption we can find a parameter regime where the experimental demonstration is possible, albeit these parameters were just out of reach with our photonics setup but can very well be achieved in the near future. We will also discuss later how Arthur may in fact be able to force this behavior of Merlin, namely instead of assuming that dishonest Merlin sends states with the correct mean photon number, Arthur can verify this himself by slightly changing the protocol itself. In any case, we emphasize again that our goal here is to define a specific theoretical scenario and a concrete computational task for which we can show a quantum advantage.

We denote the imperfect visibility of Arthur's interferometer by $\nu$ (with $\nu = 1$ in the ideal case) and the dark count probability of the single-photon detectors by $p_{\text{dark}}$. As we will justify later, the effect of the random detection events due to the dark counts can be neglected. To understand the effect of the imperfect visibility, we see that, for example, for an input state in the beam splitter at the $k$th time step $|\alpha\rangle_k \otimes |\alpha\rangle_k$ (corresponding to $x_k = 0$), the output state will be $\left|\sqrt{2\nu}\alpha\right\rangle_{D_0,k} \otimes \left|\sqrt{2(1-\nu)}\alpha\right\rangle_{D_1,k}$, hence there is a non-zero probability of a click in the wrong detector ($D_1$ in this case). We can then calculate the probability of detecting a photon in the correct and wrong detector (and nothing in the other) as follows,

$$p_c = (1 - e^{-2\nu|\alpha|^2})e^{-2(1-\nu)|\alpha|^2} \qquad (5)$$

$$p_w = (1 - e^{-2(1-\nu)|\alpha|^2})e^{-2\nu|\alpha|^2}. \qquad (6)$$

Moreover, we calculate the probability of a click in both detectors as,

$$p_{dc} = (1 - e^{-2\nu|\alpha|^2})(1 - e^{-2(1-\nu)|\alpha|^2}). \qquad (7)$$

These double clicks do not contain any information but, as we have explained, they will be used by Arthur to pick a random value for the variable, so they play a role in the verification test. Note that the average number of expected detector clicks is given by $(p_c + p_w + p_{dc})N \approx p_h N$ (with an equality for negligible $p_{\text{dark}}$ as in our case). Note also that all quantities depend on $|\alpha|^2$ and $\nu$, but we have neglected the effect of the losses in the system, as we will also justify later.

Let us now calculate, taking into account the above, the expected number of satisfied measured clauses Arthur should obtain in the YES and NO instances. In the YES instance, all clauses are satisfied by the assignment, and the probability that Arthur measures a satisfied clause will be the sum of three terms,

$$p_Y = (p_c + p_{dc}/2)^4 + (p_w + p_{dc}/2)^4 \\ + 4(p_c + p_{dc}/2)^2(p_w + p_{dc}/2)^2. \qquad (8)$$

The first term is the probability of getting four correct values for the four variables; the second of getting four wrong values; and the third is the sum of the probabilities of two correct and two wrong values in a way that the 2-out-of-4 clause remains satisfied.

In the NO instance, we upper bound the probability of measuring a satisfied clause as follows,

$$p_N \lesssim p_h^4 - \delta p_Y - (1 - \delta)(p_h^4 - p_Y). \qquad (9)$$

This is the probability of measuring a clause (for negligible $p_{\text{dark}}$) minus the probability of measuring an unsatisfied clause. To provide a bound on the latter we note that, for any assignment, there is at least a $\delta$ fraction of unsatisfiable clauses that will not be satisfied if measured correctly, namely with probability $p_Y$, and a

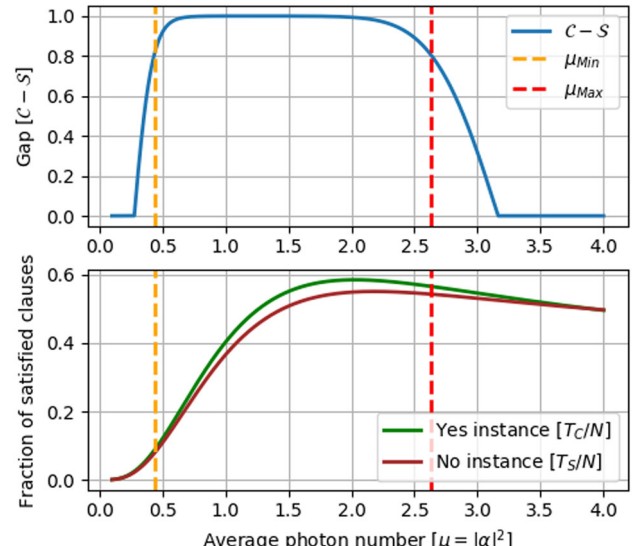

**Fig. 2 Numerical results.** (Top) Gap between completeness and soundness as a function of the mean photon number $\mu = |\alpha|^2$, for $N = 10,000$, $\delta = 0.15$, $\nu = 0.91$. The two vertical lines correspond to the minimum and maximum $\mu$ in order to have at the same time completeness $\mathcal{C} > 0.9$ and soundness $\mathcal{S} < 0.6$. (Bottom) Fraction of measured satisfied clauses as a function of $\mu$. As the mean photon number increases the number of satisfied clauses in the NO instance overcomes the one in the YES instance.

fraction $1 - \delta$ of satisfiable clauses that will be unsatisfied if measured incorrectly, namely with probability $p_h^4 - p_Y$.

It is then straightforward to find the expected number of measured satisfied clauses $T_{\mathcal{C}}$ in the YES instance and $T_{\mathcal{S}}$ in the NO instance, by multiplying the above probabilities with the number of clauses that we assume is some constant (greater than 1) times $N$. Thus, we have $T_{\mathcal{C}} - T_{\mathcal{S}} \geq (p_Y - p_N)N$. Our experimental values will be such that $T_{\mathcal{C}} - T_{\mathcal{S}}$ is a large enough number to allow us to use Chernoff bounds to guarantee a sufficiently large gap between completeness and soundness.

More specifically, we define a threshold for Arthur's verification as $T = (T_{\mathcal{C}} + T_{\mathcal{S}})/2$, in other words Arthur accepts if and only if at least $T$ measured clauses are satisfied. By a simple Chernoff bound we can then see that the completeness can go arbitrarily close to 1 and the soundness arbitrarily close to 0 by properly tuning the value of $|\alpha|^2$, and again as $|\alpha|^2 = O(N^{-1/4})$. More precisely, we use the following inequalities for completeness and soundness,

$$\mathcal{C} = \Pr[\text{correct measured clauses} \geq T]$$
$$\geq 1 - e^{-\frac{(T_{\mathcal{C}} - T_{\mathcal{S}})^2}{4T_{\mathcal{C}}}} \qquad (10)$$

$$\mathcal{S} = \Pr[\text{correct measured clauses} \geq T]$$
$$\leq e^{-\frac{(T_{\mathcal{C}} - T_{\mathcal{S}})^2}{4T_{\mathcal{S}}}}. \qquad (11)$$

To illustrate how this analysis allows us to identify an experimental parameter regime where it is possible to demonstrate a quantum advantage for our verification task, we show in Fig. 2 theoretical bounds for the fraction of measured satisfied clauses in the YES and NO instances, as well as the gap between the completeness and soundness, as a function of the mean photon number $\mu = |\alpha|^2$, for $N = 10000$, $\nu = 0.91$, $\delta = 0.15$, and negligible dark counts. We can see that for our aforementioned target gap, where we want to keep the completeness above 0.9 and the soundness below 0.6, there is a region of $\mu$ where quantum advantage can be shown for the chosen parameters.

---

**Box 1 Protocol for NP verification**

Input: Instance of the NP-complete problem and all its relevant parameters: $N$, $\delta$, etc., after the reduction to a 2-out-of-4 SAT;

Goal: Verification of the solution;

1 Merlin and Arthur jointly perform a pre-calibration of the optical setup, finding the values of the visibility $\nu_N$ and the transmittivity $\eta$;

2 Arthur computes the minimum value of the mean photon number number $\mu_N$ in order to satisfy the quantum advantage conditions 1–3 and communicates it to Merlin in order to tune the amplitude of his pulses; he also computes the threshold $T$ for accepting a proof;

3 Arthur sends a signal to Merlin to trigger the protocol;

4 Merlin encodes his proof in the phases of the pulses which are then sent to Arthur;

5 Arthur interferes Merlin's pulses with his own and assigns a value $x_k$ each time he registers a measurement in the $k$th pulse:

5.1 $x_k = 0$ for a click in detector $D_0$ and no click in $D_1$;

5.2 $x_k = 1$ for a click in detector $D_1$ and no click in $D_0$;

5.3 $x_k$ is randomly assigned if both detectors click.

6 For all the measured bits that form a clause, Arthur checks the satisfiability;

7 If the number of satisfied clauses is greater than $T$, Arthur accepts the proof, otherwise he rejects.

---

Let us now discuss the more general scenario where the dishonest Merlin may send any unentangled state (including with no or many more photons). This is a more complicated case to analyze, but we do know that whatever Merlin does, Arthur will still receive a value for a variable from the photons he inputs himself in the interferometer, which is at least $p_d = 1 - e^{-|\alpha|^2}$. If we drop the assumption that Merlin will only send coherent states with the correct mean photon number, we can still find a region with a positive gap between completeness and soundness, albeit with more stringent experimental conditions that were not fulfilled in our setup, in particular with respect to the required visibility, but that we believe can be fulfilled in the near future.

Note also that Arthur could potentially try to force Merlin to send states with the correct mean photon number by creating the pulses himself, sending them over to Merlin who prepares the state with the setup of Fig. 1 and returns it. Arthur can use random timings for his pulses impeding Merlin from injecting more photons and also use part of the pulses in order to count the number of clicks and convince himself that Merlin is not sending fewer photons over. Again, we do not need to do any of this for our demonstration of a quantum advantage, since we are free to define the computational task ourselves, namely verification of NP problems for a specific type of interactive proof systems, without having to deal with general cryptographic considerations and dishonest behaviors. Nevertheless, this would provide a simple solution in case one might want to use our protocol in practical scenarios, for example for server–client verification.

Last, we claim that losses are not important in our setting. Again, this is a verification scenario where an honest Arthur tries to efficiently verify an NP instance with the "small" help of an untrustful Merlin. Hence, Arthur and Merlin can jointly measure the potential losses during a calibration phase before the actual verification starts and increase the power of their pulses by the factor $1/\eta$, where $\eta$ includes the channel and detection efficiency. Thus, we do not have to worry here about an Arthur that can use the losses to his benefit.

To summarize the above and in preparation for the description of our experimental implementation, we provide below a step-by-step outline of the protocol in Box 1.

**Experimental results**. We now have all the ingredients to describe the experimental implementation of our verification test and the assessment of the quantum advantage for this task. As we defined previously, we need to satisfy three conditions to show quantum advantage. We need the verification procedure to take time linear in $N$, to have completeness and soundness such that $\mathcal{C} > 0.9$ and $\mathcal{S} < 0.6$, and that the number of clicks Arthur registers is much smaller than the input size $N$.

First, as we will see, in our experiment we use indeed a train of coherent pulses of size $N$ and some simple classical post-processing of the measurement results, so our test satisfies condition 1. In fact, the real time to run the verification procedure for $N$ between 5000 and 14,000 was a fraction of a second for the quantum part, a few seconds for the classical post-processing and a couple of minutes for the calibration procedure for each run.

Second, we will show that our verification procedure has high completeness, i.e., when the instance is satisfiable and Merlin sends to Arthur a satisfying assignment encoded in the coherent states, then Arthur accepts with high probability. For the same experimental parameters we will then use our theoretical analysis that upper bounds the maximum soundness of our protocol for any strategy of Merlin, and ensure that the soundness is much lower than the experimentally demonstrated completeness, thus proving condition 2 of quantum advantage.

In fact, to simplify the classical pre- and post-processing, we experimentally perform a modified version of the test, where we do not sample balanced and probabilistically checkable YES instances with planted satisfying assignments (this is far from being straightforward), but we generate uniformly random $N$-bit strings (for several values of $N$) that correspond to satisfying assignments. Note that a uniform distribution of the satisfying assignments is the hardest case for the problem, since with any other distribution, Arthur would already have some information about the possible solutions to the problem. After that, we check the number of the variables for which Arthur obtains the correct value, the number of wrong values, and the number of undefined variables. From these numbers we compute the expected number of satisfied and unsatisfied clauses Arthur will get on a random YES instance, and using the threshold that has been defined in the calibration phase of the experiment described below, we conclude whether Arthur would accept or reject the instance, thus estimating the completeness of our protocol.

Finally, the measurements events of Arthur are also used to ensure that condition 3 for quantum advantage is satisfied.

Let us now provide more details on our experiments. The experimental setup is shown in Fig. 3. The coherent light pulses are generated using a continuous wave laser source emitting light at 1560 nm followed by an amplitude modulator, at a rate of 50 kHz and with a pulse duration of 10 ns. An unbalanced beam splitter is used to monitor the pulse power and a variable optical attenuator (VOA) to set the mean photon number at the desired level. We then use a balanced beam splitter to direct the coherent pulses to Arthur and Merlin. Following the scheme for the verification test shown in Fig. 1, Merlin impinges his proof on the phase of the pulses using a phase modulator (PM). Arthur and Merlin then both use a set of variable optical attenuators to finely tune and equalize the power of the signals entering the output

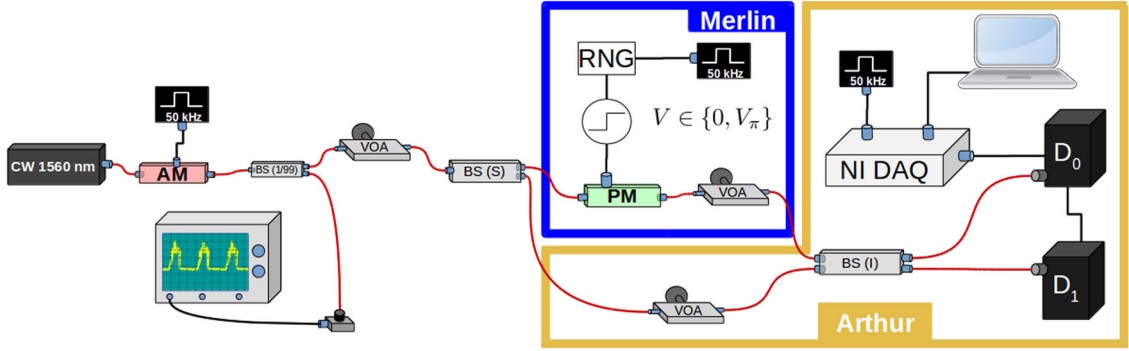

**Fig. 3 Experimental setup.** A coherent light source operating at a wavelength of 1560 nm (Pure Photonics) together with an amplitude modulator (AM) are used to generate coherent pulses at a 50 kHz repetition rate and with a 10 ns pulse duration. Using a beam splitter with 1/99 ratio, we monitor the pulse power with a photodiode and send the small fraction of the beam to the rest of the setup. The beam is further attenuated before being split with a balanced beam splitter (BS) and sent to Merlin and Arthur. The former encodes the proof in the phases of his pulses using a phase modulator (PM). They both use attenuators to fine tune and equalize the photon number in their paths and the pulses are then interfered on the output beam splitter (I) before been detected by InGaAs avalanche photodiode single-photon detectors (IDQuantique). The measurement outcomes are collected using a National Instruments data acquisition card and analyzed with dedicated software.

balanced beam splitter of the interferometer. The pulses are finally detected by two InGaAs single-photon detectors ($D_0$ and $D_1$) and the measurement results are collected by Arthur. The experiment is controlled by a data acquisition card and the data is analyzed with dedicated software.

We perform several preliminary measurements and calibrations before moving on with the verification test. In particular, we calibrate the voltage level needed to induce a $\pi$-phase shift, $V_\pi$ with the phase modulator off line. Phase drifts may occur during the experiment and affect the obtained visibility, hence requiring real-time phase correction techniques[27]. In our case the time scale of the drift (on the order of 5 s) was much longer than the duration of each run of the protocol (around a fraction of a second) and it was therefore not necessary to use such feedback loops. Arthur and Merlin also need to carefully equalize the power of their pulses before interfering them, as required by our test. To do this, Arthur calibrates the losses in Merlin's path by first removing his signal, measuring detection events due to Merlin's signal only, for several values of the mean photon number, and then minimizing the clicks on one of the detectors with his signal reconnected. This procedure also allows Arthur and Merlin to determine the losses in their setup, and hence the efficiency $\eta$, which includes the channel efficiency $\eta_{channel} \approx 38\%$, and the quantum efficiency of the single-photon detectors, $\eta_{det} \approx 25\%$. As we have explained, this parameter does not play a direct role in our verification test.

Importantly, the above calibration procedure allows Arthur to evaluate the visibility of the interferometer, which is central to the assessment of the performance of our test. Indeed, we use this estimation as benchmark for the expected number of satisfied clauses in the YES and NO instances, and correspondingly define a threshold for accepting a proof, as we have detailed previously. A low visibility will increase the number of errors so that we will need to increase $\delta$ in order to verify the solution with sufficient completeness and soundness.

In our experiment, we use the nominal value $\nu_N = 0.93$, as well as $\mu_N = 1.31$, and set correspondingly $\delta = 0.15$. These values are chosen such that in our theoretical estimations (see Fig. 2) the conditions $\mathcal{C} > 0.9$ and $\mathcal{S} < 0.6$ are satisfied at the same time for all the values of $N$ that we will be using. The value of $\delta$ will be fixed for all the runs; however, we experimentally measure the actual visibility in each case. We remark that here we are using a single laser to generate the pulse sequences of Arthur and Merlin, which is optimal for obtaining high visibility values. Nevertheless, it is still possible to use this setup for assessing the performance of our test

for demonstrating a quantum advantage since all actions required by the test, as shown in Fig. 1, are performed independently.

We finally remark that the dark count probability in our setup is $p_{dark} \sim 10^{-3}$, and hence the effect of dark counts can safely be considered negligible for our values of $\nu$ and $\mu$. In fact, for our choice of parameters, we have $p_c, p_w, p_{dc} \gtrsim 10^{-2}$ as can be easily seen from Eqs. (5) and (6).

We are now ready to analyze our verification test enabling Arthur to verify efficiently that a given 2-out-of-4 SAT instance is satisfiable. As we have explained, we assume that Merlin acts honestly and only the environment will lead to errors that will make Arthur reject a correct proof. After performing the preliminary calibrations, Merlin starts the test by encoding his proof on his coherent pulse sequence. Here, as a proof, we generated a random Boolean string of $N$ variables (for several values of $N$). Arthur records all clicks $t_{clk}$ including single and double clicks on both detectors. We denote the single clicks as $s_{clk}$. He assigns a bit 0 or 1 to variable $x_k$ if the pulse at time step $k$ resulted in a single click in detector $D_0$ or in a single click in detector $D_1$, respectively. For the double clicks, he assigns a random value to the corresponding variable, while we leave all other variables undefined.

For computing the completeness of the verification, we need to decide if Arthur would have accepted or rejected the specific run of the verification test. Had we fixed a specific instance then Arthur would just check with the values of the variables that he has obtained, how many clauses are satisfied and how many clauses are not, and depending on the threshold $T$ he would accept or reject. Note that Arthur can indeed compute the value of $T$ given the experimental values of $\mu$ and $\nu$.

As we said, in order to avoid the complications of sampling such classical instances in a fair way, we decide whether Arthur accepts or rejects the instance using the same threshold $T$, but estimating the number of clauses Arthur would have found satisfied or not, through the number of correct variable values he really obtained through the experiment. Since the instances are assumed to be balanced, this is equal on expectation over random instances to the corresponding calculations on the clauses.

In other words, from the number of all single clicks $s_{clk}$, the number of single clicks that correspond to the correct variable value $c_{clk}$, and the number of double clicks that are randomly assigned $dc_{clk}$, we can infer the probabilities $p_{dc_{exp}} = \frac{t_{clk}}{N}, p_{c_{exp}} = \frac{c_{clk}}{N}$ and $p_{w_{exp}} = \frac{s_{clk} - c_{clk}}{N}$, from which we can compute the expected

number of satisfied clauses in the YES and NO instances using Eqs. (8) and (9). Note that the expected numbers are sufficiently far from the threshold so that we do not expect the variance of the number of satisfied clauses (for each specific instance) to affect the completeness. For these experimental parameters we also compute the soundness, which is in fact very close to 0, see Fig. 4.

In order to prove the third condition for the quantum advantage, if the proof is accepted, we count the number of variables for which Arthur has no information, i.e., $N - s_{clk}$,

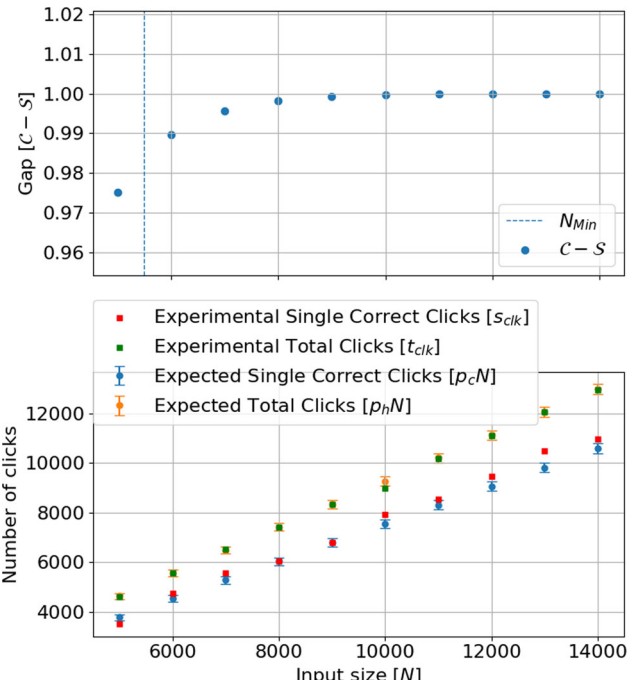

**Fig. 4 Experimental data.** (Top) Plot of the gap as a function of $N$ when simulating the protocol with the nominal parameters of $\nu_N = 0.93$, $\mu_N = 1.31$ and $\delta = 0.15$. The vertical line bounds the region for quantum advantage. (Bottom) Number of clicks as a function of $N$. The correct bits are clicks in the correct detector or in both detectors with half probability and total clicks is the total number of measured pulses. Each square corresponds to one run of the protocol whereas the dots with error bars are numerical. Because each pulse gives a poissonian probability distribution in the number of photons, the error bar is given by $2\sqrt{\#\,\text{clks}}$ which is twice the root mean square of the poissonian.

which is the information that Arthur is missing to complete the solution. We remark again that a double click in both detectors does not provide any information to Arthur and we also assume that all single clicks reveal the true variable value. With only classical resources, Arthur would need a computational time of $2^{\gamma(N - s_{clk})}$, for some prefactor $\gamma$ (for SAT solvers around 0.4). As we have explained, here we claim quantum advantage if $N - s_{clk}$ is larger than 1000, but it is clear that for any given threshold one can reach quantum advantage by increasing $N$ and improving $v$.

In Table 1 we summarize our experimental data for fixed $\delta$, slightly varying $\mu$, and $v$ evaluated for every input size $N$. We include the number of single clicks, correct clicks, double clicks, missing bits, as well as the threshold $T$ and the number of computed satisfied clauses in each case. As we can see, the number of bits Arthur still misses at the end of the protocol increases with $N$, which means that the problem is becoming more and more difficult for classical computation as $N$ increases. Moreover, starting from $N = 6000$, we see that the computed number of satisfied clauses is much bigger than the threshold, hence the completeness is very close to one.

Finally, in Fig. 4 we compare the simulations with a typical run of the experiment for various $N$ fixing the nominal photon number $\mu_N$, visibility $\nu_N$ and the constant $\delta$. Notice how the gap between completeness and soundness increases with $N$ and very fast becomes almost 1. In the experimental runs shown in the figure, the only point for which we cannot show quantum advantage is the one at $N = 5000$, since the gap between completeness and soundness is not large enough. This is due to a low level of visibility that induced a too large number of incorrect detections in this case.

## Discussion

Our result is an experimental demonstration of a computational quantum advantage in the interactive setting with linear optics. The simplicity of our experimental implementation, in addition to the powerful algorithmic idea of the Sampling Matching, exemplifies the power of linear optics, and in particular of coherent state mappings, not only for communication but also for computational tasks. It will be interesting to investigate further applications of linear optics, in particular in the frame of near-term quantum technologies. Moreover, we would like to argue that our computational task, that of efficiently verifying NP-complete problems with limited leakage of knowledge about the proof, is a step closer to useful applications, even though it remains for the time being a theoretical scenario. In fact, one can start imagining applications in a near-term quantum cloud, where

**Table 1 Summary of experimental data. In each run we increase the input size N by 1000. The table shows: the actual visibility in each run; the average number of photons per pulse; the number of measured single clicks and those that were in the correct detector; number of double clicks, which correspond to randomly assigned variables; the missing bits to complete the solution; the threshold of correct measured clauses for accepting a proof; the number of satisfied clauses in the experiment. The parameters $\delta = 0.15$, $\nu_N = 0.93$, and $\mu_N = 1.31$ are kept fixed in the theoretical analysis of the experiment.**

| N | $\nu$ | $\mu$ | Total single clicks | Correct clicks | Double clicks | Missing bits | Threshold | Satisfied clauses |
|---|---|---|---|---|---|---|---|---|
| 5000 | 0.87 | 1.29 | 3657 | 3505 | 964 | 1343 | 2254 | 2227 |
| 6000 | 0.93 | 1.30 | 4834 | 4741 | 719 | 1166 | 2717 | 3231 |
| 7000 | 0.94 | 1.34 | 5670 | 5582 | 848 | 1330 | 3232 | 3904 |
| 8000 | 0.92 | 1.29 | 6203 | 6062 | 1195 | 1797 | 3613 | 4030 |
| 9000 | 0.92 | 1.30 | 6974 | 6813 | 1363 | 2026 | 4088 | 4546 |
| 10,000 | 0.95 | 1.15 | 8045 | 7929 | 947 | 1955 | 4111 | 5082 |
| 11,000 | 0.93 | 1.30 | 8675 | 8524 | 1515 | 2325 | 4996 | 5789 |
| 12,000 | 0.93 | 1.30 | 9632 | 9466 | 1476 | 2368 | 5437 | 6471 |
| 13,000 | 0.95 | 1.30 | 10636 | 10496 | 1405 | 2364 | 5902 | 7320 |
| 14,000 | 0.94 | 1.29 | 11135 | 10950 | 1807 | 2865 | 6801 | 7437 |

a powerful quantum server might have the ability to perform some difficult computation, and the much less powerful client can verify the validity of the computation, without the server needing to reveal all the information to the client. Such limited-knowledge proof systems could also have applications in a future quantum internet, similarly to classical zero-knowledge proofs that can be used for identification, authentication or blockchain. It still remains an open question to find the first concrete real-world application of quantum computers and our results show that linear optics might provide an alternative route towards that goal.

## Data availability

The experimental data that support the finding are available from the authors upon request.

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

## Acknowledgements

We thank Matteo Schiavon for useful discussions and practical advice. We acknowledge financial support from the European Research Council project QUSCO (E.D.) and the French National Research Agency project quBIC.

## Author contributions

F.C. operated the experimental setup and analyzed the data. N.K. contributed to the design of the experiment. All authors contributed to the theoretical analysis and to writing the manuscript. E.D. and I.K. conceived and supervised the project.

## Competing interests

The authors declare no competing interests.
