## [Peer Review File · Nature Communications]

Reviewers' Comments:

Reviewer #1:

Remarks to the Author:

The present manuscript reports an experimental implementation of a linear optics apparatus designed to demonstrate a computational quantum advantage in a specific task. More specifically, such quantum advantage is reported for the task of verifying the solution to a NP-complete problem (2-out-of-4 SAT) when only partial information on the solution is available.

The main idea behind the scenario is not entirely new, since it is constructed on former papers (mainly, Refs. [25,26]). The main novelty of the manuscript with respect to previous papers (in particular with Ref. [25]) is represented by the identification of a different linear optical system to implement the verification strategy, which based on temporal encoding in coherent states. The authors then present a detailed analysis of the protocol, provide a benchmark with classical verification, and finally report an experimental implementation reaching the quantum advantage scenario.

Even if the problem itself addressed by this manuscript is not new, my opinion is that this paper represent a relevant contribution to the quantum computation framework. Indeed, while the first quantum computational supremacy implementation has been reported last year, this paper addresses such scenario by addressing a different task, which may potentially have several applications, and a different platform. Thus, I would be supportive towards its publication in Nature Communications.

I have a very few comments, enlisted below, that I would like the authors to address.

(1) Some details of the verification protocol, at least to my understanding, are not entirely clear. More specifically, I am referring to how the clauses are verified after the measurement process (Eq. 4). I would ask the authors to add more details on this aspect.

(2) The authors perform a detailed analysis on the role of experimental imperfections, and how to deal with them by defining appropriate thresholds for the verification process. At least to my understanding, all this scenario is based on the assumption the Merlin acts honestly, thus sending the coherent state with the appropriate amplitude given the noise estimation performed a priori. Even if I understand that the authors make this assumption given the "freedom" to choose the computational problem, it seems to me that this assumption significantly limits the range of applications of the approach. More specifically, most applications directed towards cloud and blind quantum computing require to not make such "trustness" assumption on part of the nodes, and in certain cases are oriented towards identifying whether the nodes are acting not honestly. I would like the authors to provide a more detailed comment on such aspect in the manuscript.

Reviewer #2:

Remarks to the Author:

In this paper the authors claim to have made the first experimental demonstration of a computational quantum advantage with linear optics. I do not believe such a statement can be made without significant caveats and hence the claim in this manuscript should be reworded and the manuscript revised to more accurately reflect the positioning of this result.

The term quantum computational advantage has become broadly understood as performing a task on a quantum device that is practically infeasible for a classical device. This division is made using minimal assumptions, usually just those about the standard division of complexity classes based on polynomial vs exponential separations.

An NP-complete problem, by definition is deterministically and efficiently verifiable using only classical resources. The protocol in this paper claim a quantum advantage to verifying NP-complete problems. To obtain a sense of quantum advantage, the process described in this paper requires a verification procedure where the verification certificate (or proof) is substantially curtailed (of length linear in N). This is not a practical consideration but one of theoretical interest in the analysis of these protocols.

Therefore, I believe the claims made in this paper need to be significantly changed to reflect that this result is based on a particular theoretical scenario and is not statement on the standard interpretation of quantum computational advantage. For if merely adding quantum resources to gain any advantage in a particular chosen scenario is the definition of quantum computational advantage, then quantum computational advantages have been seen for a long time, even including all the linear optics experimental platforms that have been performing experiments over the decades.

Nevertheless, the major step in the paper seems to be that single photon experiments of [26] are replaced with coherent states. This is a very natural choice as the original scheme needed repeated copies of single photon experiments and coherent state displacements are generated by a single annihilation operator. There are some details of doing this on how to achieve the measurement of the bitstream data. But, the techniques used are essentially an unambiguous discrimination of two coherent states which has been known for many decades. In the end, the details of this technique aren't super important as the coherent state amplitudes are "tuned" in the experiment and if the values used were only fractionally larger, then standard homodyne detection would provide a very good measurement of the encoded data. Coherent states are also robust to losses due to their lack of entangling power, so the protocol's robustness to loss is somewhat expected. The particular novelty of the scheme that goes beyond bringing coherent states to single photon experiment needs to be emphasised more in the paper.

Reviewer #3:

Remarks to the Author:

This paper claims a quantum advantage for *verifying* the solutions to NP-complete problems, using an optical variant on a QMA(2) protocol that was proposed in 2008 by Aaronson, Bouland, Drucker, Fefferman, and Shor (ABDFS). There are two parts of the paper, theoretical and experimental. Unfortunately, I'm not competent to judge the details of the experiment, so I'll confine myself to comments that don't depend on that.

It's interesting that the ABDFS protocol can be implemented using coherent states, and especially interesting if the authors managed to implement this with thousands of photons (!). Having said that, I'm worried that, if this paper is published in its current form, it will predictably give rise to wild misinterpretations: for example that there's an exponential speedup here for *solving* NP-complete problems, or even an exponential advantage for verifying their answers (as opposed to a small polynomial advantage in the size of the witness, under an assumption of unentanglement). I'm also worried that basic questions seem hard to find answers to in the manuscript, and that overly dramatic claims are made. Here are my main comments:

- The entire framing, that QC solves a problem in minutes that would take longer than the age of the universe classically, relies on a way of defining the problem that no one in computer science would ever use. Namely, there's a polynomial reduction in the length of a witness, from N bits to $\sim N^{\{3/4\}}$ bits (although that relies on a further assumption of unentanglement). But instead of presenting this as a polynomial improvement, the authors fix the size of the witness at $\sim N^{\{3/4\}}$ (i.e., exactly what their quantum protocol achieves). They then stipulate that both classical and quantum provers need to provide witnesses of exactly that length -- and if the classical prover can't fit its witness into that length, the verifier then needs to perform a brute-force search to

complete the rest of the witness, which takes $\exp(N - O(N^{3/4}))$ time. This is the basis for their claim of an "exponential speedup." The trouble is that, by these rules, pretty much *any* polynomial advantage could be claimed to be "exponential" -- one simply needs to set a cutoff at the smaller polynomial, after which an exponential search is needed because of the way the problem is defined.

Furthermore, for readers who aren't expert in CS, framing things in this tendentious way could encourage precisely the misconception one is worried about, that this in some shape or form represents an "exponential speedup for solving NP-complete problems" (and an experimentally demonstrated one, no less!).

So I think the entire "exponential speedup" framing needs to be eliminated from the paper. The headline -- in the abstract, the intro, and everywhere else -- should be the reduction that's achieved in the *number of bits of witness*, both theoretically and experimentally. It should not be about doing something (i.e., finding a witness given $\sim N^{3/4}$ helper bits) in a few minutes that classically takes exponential time.

- Also, as mentioned earlier, the protocol, derived from ABDFS'08, only works if the witnesses are unentangled. And, in practice, one would normally have no physical way to verify that witnesses were unentangled. This is a pretty crucial point, enough that I think it needs to be mentioned in the abstract.

- The ABDFS protocol needs witnesses with only $\sim O(\sqrt{N})$ qubits total, while this paper needs $\sim O(N^{3/4})$. I'm sure it's there somewhere, but I looked several times and was unable to find a clear statement of why the \sqrt{N} from ABDFS becomes $N^{3/4}$ here. Is it because of the constraints of an optical implementation? This ought to be possible to explain in a couple sentences early on.

- Surprisingly, it was hard to find a clear statement in the paper of *basic numbers*: how many variables did the SAT instances have? How many photons were there in the optical setup? There are tables from which I infer that the answers are "thousands," but whatever the answers are, they should be in the abstract and intro, and one shouldn't need to parse data tables to get them!

- Also, how were the SAT instances generated? Were the clauses chosen randomly, or what? Again: obvious question, I'm sure it's answered somewhere, but I could not find the answer after multiple rereadings.

- And a related point: the statements about the classical time needed to solve the SAT instances seem to involve just taking the best-known worst-case exponential upper bound for 3SAT, and then applying it "off the shelf." But if one is interested in practical comparisons, this is absurd: one needs to look at state-of-the-art SAT-solving heuristics, and evaluate their performance in practice. Was this done? If not, then it's even more reason to frame things in terms of the witness size instead of running time.

Anyway, while the paper *cannot* be accepted in its current form, I actually think it's interesting and could be suitable for Nature Communications, if (1) the above points are satisfactorily addressed and (2) the more experimentally-inclined reviewers agree.

Nature Communications - Reply to Reviewer comments on “Experimental demonstration of quantum advantage for NP-verification”

November 19, 2020

1 Reviewer #1:

The present manuscript reports an experimental implementation of a linear optics apparatus designed to demonstrate a computational quantum advantage in a specific task. More specifically, such quantum advantage is reported for the task of verifying the solution to a NP-complete problem (2-out-of-4 SAT) when only partial information on the solution is available.

The main idea behind the scenario is not entirely new, since it is constructed on former papers (mainly, Refs. [25,26]). The main novelty of the manuscript with respect to previous papers (in particular with Ref. [25]) is represented by the identification of a different linear optical system to implement the verification strategy, which based on temporal encoding in coherent states. The authors then present a detailed analysis of the protocol, provide a benchmark with classical verification, and finally report an experimental implementation reaching the quantum advantage scenario.

Even if the problem itself addressed by this manuscript is not new, my opinion is that this paper represent a relevant contribution to the quantum computation framework. Indeed, while the first quantum computational supremacy implementation has been reported last year, this paper addresses such scenario by addressing a different task, which may potentially have several applications, and a different platform. Thus, I would be supportive towards its publication in Nature Communications.

We thank the Reviewer for their positive assessment of our work.

I have a very few comments, enlisted below, that I would like the authors to address.

(1) Some details of the verification protocol, at least to my understanding, are not entirely clear. More specifically, I am referring to how the clauses are verified after the measurement process (Eq. 4). I would ask the authors to add more details on this aspect.

We have made the description of the protocol clearer and added more details in the paper, in particular in the section following Eq. (4). In short, Arthur assigns values to the variables in the following way: 0, if there is a single click on detector D0; 1 if there is a single click on detector D1; a random value 0 or 1, if there is a double click; no assigned value if there is no click. Then given the values of the variables, Arthur will look at all the clauses of the instance and for the ones where all four variables have been assigned values, he will check whether the clause is satisfied or not. In the end, given the number of satisfied and unsatisfied clauses, he will accept or reject.

(2) The authors perform a detailed analysis on the role of experimental imperfections, and how to deal with them by defining appropriate thresholds for the verification process. At least to my understanding, all this scenario is based on the assumption the Merlin acts honestly, thus sending the coherent state with the appropriate amplitude given the noise estimation performed a priori. Even if I understand that the authors make this assumption given the “freedom” to choose the computational problem, it seems to me that this assumption significantly limits the range of applications of the approach. More specifically, most applications directed towards cloud and blind quantum computing require to not make such “trustness” assumption on part of the nodes, and in certain cases are oriented towards identifying whether the nodes are acting not honestly. I would like the authors to provide a more detailed comment on such aspect in the manuscript.

In the theoretical part of the paper, the analysis of the protocol is performed with the assumption of a dishonest Merlin, who still cannot, however, entangle the quantum proofs. Getting rid of the unentanglement assumption seems to be impossible as discussed in Ref. [25]. In the experimental implementation of the protocol, the visibility of our interferometer was slightly below the threshold to ensure a positive gap between completeness and soundness

without further assumptions on Merlin's behaviour. As a consequence, and in order to find parameter regimes where we could show in practice our protocol, we assumed that Merlin can only cheat in a more restricted "classical" way, i.e., he cannot generate an arbitrary quantum state but can only dishonestly choose an assignment to the variables and apply a phase to the coherent pulses. In the paper we briefly discuss how Arthur could enforce this behaviour on Merlin's side, for example using random timings for his pulses, hence making the protocol work against any dishonest Merlin, and enabling for example server-client quantum computing. We note however that this is not an inherent limitation and it will be possible to enhance our proof-of-principle experiment with even better experimental components allowing for an implementation against any dishonest Merlin.

2 Reviewer #2:

In this paper the authors claim to have made the first experimental demonstration of a computational quantum advantage with linear optics. I do not believe such a statement can be made without significant caveats and hence the claim in this manuscript should be reworded and the manuscript revised to more accurately reflect the positioning of this result.

We thank the Reviewer for their comments that have helped to considerably improve the manuscript. Following the Reviewer's recommendation, we have reworded our statements and made our claims clearer in the abstract, the introduction and in several places in the rest of the paper.

The term quantum computational advantage has become broadly understood as performing a task on a quantum device that is practically infeasible for a classical device. This division is made using minimal assumptions, usually just those about the standard division of complexity classes based on polynomial vs exponential separations.

An NP-complete problem, by definition is deterministically and efficiently verifiable using only classical resources. The protocol in this paper claim a quantum advantage to verifying NP-complete problems. To obtain a sense of quantum advantage, the process described in this paper requires a verification procedure where the verification certificate (or proof) is substantially curtailed (of length linear in N). This is not a practical consideration but one of theoretical interest in the analysis of these protocols.

Therefore, I believe the claims made in this paper need to be significantly changed to reflect that this result is based on a particular theoretical scenario and is not statement on the standard interpretation of quantum computational advantage. For if merely adding quantum resources to gain any advantage in a particular chosen scenario is the definition of quantum computational advantage, then quantum computational advantages have been seen for a long time, even including all the linear optics experimental platforms that have been performing experiments over the decades.

We have reworded our claims in the revised version of the paper in order to take into account the Reviewer's comments. Let us discuss them in more detail :

First, we agree with the Reviewer that our problem is primarily of theoretical interest, similarly to the other problems that have been used before in this setting, such as Boson Sampling or random circuit sampling. We therefore agree that our result does not correspond to a specific application in a practical sense but to a theoretical scenario where there exists a task that a quantum computer can perform fast, while a classical computer needs extensive amounts of time. We would still like to point out, however, that the potential practical interest of curtailing the witness should not be neglected. Indeed, we would like to emphasize that the case of verification of NP-complete problems without revealing all or part of the witness is something that has been theoretically studied before extensively in what is called Zero-Knowledge proofs (see for example "Leakage-Resilient Zero Knowledge", Sanjam Garg, Abhishek Jain, and Amit Sahai, CRYPTO 2011). Again, this is not to claim that our advantage is for a practically relevant task, but that it can potentially find applications, same as the random circuit sampling or Boson Sampling can as well.

Next, in order to make the difference with what the Reviewer calls the "standard interpretation of quantum computational advantage" and our setting, we named our advantage "quantum computational advantage in the interactive setting", to clarify that in our theoretical scenario, a form of interaction between two parties is allowed before the computational task is performed. Regarding the necessary assumptions, let us point out that, in fact, the computational assumptions used both for random circuit sampling and for Boson Sampling are not so minimal or well understood. This is attested by the better classical algorithms that continue to appear for these problems, see for example Ref. [2] in our paper. Our computational assumption is the exponential time hypothesis, in other words the hypothesis that NP-complete problems do not have polynomial time classical algorithms, which is widely

accepted.

Finally, regarding the computational advantages and linear optics, we totally agree with the Reviewer that linear optics has provided many advantages before. For example, photonic entanglement enables Bell's violations, and quantum communication enables QKD and also solving problems with exponentially smaller messages than classically possible in the communication complexity scenario; the authors have also contributed to such demonstrations. While all these are significant forms of quantum advantage, we would like to make the distinction with what is called a computational advantage, which as the Reviewer said, pertains to a scenario where a particular task can be done fast with a quantum computer and slow with any classical computer. This is not the case in the previous examples, where no matter how much time one can give to the classical machine, it cannot violate Bell's inequalities or perform QKD with classical communication and any amount of time. In some sense, these advantages are stronger, since they talk about the impossibility of performing something classically, but for a computational advantage we need to find a scenario where both the classical and quantum computer can perform the task, albeit in very different time. Our task achieves just that: it provides a theoretical scenario and a task, which a quantum Arthur can solve fast, while any classical Arthur would have to spend a lot of time until he can solve it.

Nevertheless, the major step in the paper seems to be that single photon experiments of [26] are replaced with coherent states. This is a very natural choice as the original scheme needed repeated copies of single photon experiments and coherent state displacements are generated by a single annihilation operator. There are some details of doing this on how to achieve the measurement of the bitstream data. But, the techniques used are essentially an unambiguous discrimination of two coherent states which has been known for many decades. In the end, the details of this technique aren't super important as the coherent state amplitudes are "tuned" in the experiment and if the values used were only fractionally larger, then standard homodyne detection would provide a very good measurement of the encoded data. Coherent states are also robust to losses due to their lack of entangling power, so the protocol's robustness to loss is somewhat expected. The particular novelty of the scheme that goes beyond bringing coherent states to single photon experiment needs to be emphasised more in the paper.

Following the Reviewer's recommendation, we have now clarified in the paper the novelty of our protocol and all the challenges we had to overcome in order to make an experimental demonstration possible. We provide below some remarks.

The single photon version of the protocol described in Ref. [26] is an important theoretical contribution, but it is considerably out of reach of any realistic implementation with state-of-the-art technology. The necessity of hundreds of copies of identical photons in a superposition state of thousands of modes, as well as a vast number of optical elements that grows with the input size of the chosen problem, make any physical implementation unthinkable. Furthermore, although the main result of Ref. [25] is that the soundness probability of the protocol is constant with N , this probability is actually really tiny if one computes it explicitly (this calculation is not provided by the authors). As a consequence, the single photon version was unusable from many points of view. More importantly, the natural choice of using the coherent state mapping proposed in Ref. [29] actually will not be sufficient to provide any parameter regime where the experiment could be performed, neither now nor in the future.

One of the main differences that made the new protocol implementable is the use of the Sampling Matching technique, where Merlin's trains of coherent state pulses are not interfered with themselves, but Arthur creates his own train of pulses to use in the interferometer. This allows us to prove that any variable has at least some probability of being detected, which was not the case in Ref. [26]. This conceptual difference is also the one that allows us to "mix" the three tests into one and thus get to a protocol that could be implemented.

3 Reviewer #3:

This paper claims a quantum advantage for *verifying* the solutions to NP-complete problems, using an optical variant on a QMA(2) protocol that was proposed in 2008 by Aaronson, Bouland, Drucker, Fefferman, and Shor (ABDFS). There are two parts of the paper, theoretical and experimental. Unfortunately, I'm not competent to judge the details of the experiment, so I'll confine myself to comments that don't depend on that.

It's interesting that the ABDFS protocol can be implemented using coherent states, and especially interesting if the authors managed to implement this with thousands of photons (!). Having said that, I'm worried that, if this paper is published in its current form, it will predictably give rise to wild misinterpretations: for example that there's an exponential speedup here for *solving* NP-complete problems, or even an exponential advantage for verifying their answers (as opposed to a small polynomial advantage in the size of the witness, under an assumption

of unentanglement). I'm also worried that basic questions seem hard to find answers to in the manuscript, and that overly dramatic claims are made. Here are my main comments:

We thank the Reviewer for their comments that have helped to considerably improve the manuscript. We have proceeded to make many clarifications in the abstract, introduction and several other places in the paper in order to avoid any misinterpretations. We provide specific remarks below.

- The entire framing, that QC solves a problem in minutes that would take longer than the age of the universe classically, relies on a way of defining the problem that no one in computer science would ever use. Namely, there's a polynomial reduction in the length of a witness, from N bits to $N^{3/4}$ bits (although that relies on a further assumption of unentanglement). But instead of presenting this as a polynomial improvement, the authors fix the size of the witness at $4 N^{3/4}$ (i.e., exactly what their quantum protocol achieves). They then stipulate that both classical and quantum provers need to provide witnesses of exactly that length – and if the classical prover can't fit its witness into that length, the verifier then needs to perform a brute-force search to complete the rest of the witness, which takes $\exp(N - O(N^{3/4}))$ time. This is the basis for their claim of an "exponential speedup." The trouble is that, by these rules, pretty much *any* polynomial advantage could be claimed to be "exponential" – one simply needs to set a cutoff at the smaller polynomial, after which an exponential search is needed because of the way the problem is defined.

Let us try to be more precise about our claim. In our scenario, there are two resources, the length of the witness revealed, and the time spent on the verification. The Reviewer prefers to fix one of them, the time, to polynomial and ask: If we fix the time to be polynomial, what is the size of the witness that needs to be revealed in the quantum and in the classical case? This is a totally legitimate question. In the same way, we fix the other parameter, the length of the witness, to sublinear, and ask: If we fix the length of the revealed witness to sublinear, what is the time needed for the verification? We also believe that this is an equally legitimate question. We have added the Reviewer's suggestion as an alternative way of seeing our result. Here is an example of the same phenomenon, where given two resources one can decide to phrase the question any two ways. For the pointer-jumping problem in communication complexity, one can define two resources: number of rounds, and length of communication. One can ask, fixing the communication to be logarithmic, how many rounds do I need for the k -level pointer jumping problem? And the answer is k . But one can also ask the following question: if I fix the number of rounds to be $k - 1$ or k , how much communication do I need to solve the problem? And then one can claim an exponential separation between the two cases, where with k rounds, the communication is logarithmic, while with $k - 1$ it is polynomial in the input size (see "Rounds in Communication Complexity Revisited", Noam Nisan and Avi Ginzerson, SIAM J. Comput., 22(1), 211–219, 1993). There are more such examples.

Furthermore, for readers who aren't expert in CS, framing things in this tendentious way could encourage precisely the misconception one is worried about, that this in some shape or form represents an "exponential speedup for solving NP-complete problems" (and an experimentally demonstrated one, no less!).

So I think the entire "exponential speedup" framing needs to be eliminated from the paper. The headline – in the abstract, the intro, and everywhere else – should be the reduction that's achieved in the **number of bits of witness**, both theoretically and experimentally. It should not be about doing something (i.e., finding a witness given $N^{3/4}$ helper bits) in a few minutes that classically takes exponential time.

We agree with the Reviewer that it is very important that one does not get the idea from the paper that quantum computers can solve NP-complete problems in polynomial time. We had already paid particular attention to this point with explicit sentences in the first version of the paper but we have re-iterated this throughout the revised version. However, we do not think that this delegitimizes the way we phrase our question, which again just fixes one of the two resources and asks about the other one.

In the revised version of the paper we explicitly and carefully explain the purpose of the protocol (verification is in the title), our definition of quantum advantage and what we have obtained. We have in particular carefully rewritten the abstract and the introduction and we believe this minimizes the possibility of any misunderstanding.

- Also, as mentioned earlier, the protocol, derived from ABDFS'08, only works if the witnesses are unentangled. And, in practice, one would normally have no physical way to verify that witnesses were unentangled. This is a pretty crucial point, enough that I think it needs to be mentioned in the abstract.

We have changed the abstract to include this fact.

- The ABDFS protocol needs witnesses with only $O(\sqrt{N})$ qubits total, while this paper needs $O(N^{3/4})$. I'm sure it's there somewhere, but I looked several times and was unable to find a clear statement of why the \sqrt{N} from ABDFS becomes $N^{3/4}$ here. Is it because of the constraints of an optical implementation? This ought to be possible to explain in a couple sentences early on.

We have provided an explicit explanation in the revised paper. In short, in ABDFS one can choose a specific measurement (depending on the clauses) to always get the value of a clause and hence check satisfiability. In fact the \sqrt{N} is needed to prove the uniformity of the state. In our case, we cannot perform such measurements (not in the lab at least!) so the way to have a good probability of measuring all variables in a clause of the instance is by measuring enough random variables, and that number needs to be now $N^{3/4}$ to make the probabilities work out. In the section “Verification test”, subsection “Completeness” of the manuscript, we have added more explanation on where this comes from.

- Surprisingly, it was hard to find a clear statement in the paper of *basic numbers*: how many variables did the SAT instances have? How many photons were there in the optical setup? There are tables from which I infer that the answers are ”thousands,” but whatever the answers are, they should be in the abstract and intro, and one shouldn’t need to parse data tables to get them!

This information is included in the experimental part of the paper. We have now provided some of these basic numbers at the end of the introduction so that one can have access to them early on in the paper.

- Also, how were the SAT instances generated? Were the clauses chosen randomly, or what? Again: obvious question, I’m sure it’s answered somewhere, but I could not find the answer after multiple rereadings.

This information is provided in detail at the beginning of the experimental section, right before going into the actual description of the experiment. We have kept it there as it makes the text more coherent and it is quite difficult to explain the details earlier in the paper.

- And a related point: the statements about the classical time needed to solve the SAT instances seem to involve just taking the best-known worst-case exponential upper bound for 3SAT, and then applying it ”off the shelf.” But if one is interested in practical comparisons, this is absurd: one needs to look at state-of-the-art SAT-solving heuristics, and evaluate their performance in practice. Was this done? If not, then it’s even more reason to frame things in terms of the witness size instead of running time.

We totally agree with the Reviewer that this would be absurd and that is why this is not what we have done. It is well known that the best-known worst case exponential upper bound for 3SAT, in fact, is simply 2^N , but this is not the number we used for the classical case, we used the best-known classical heuristic for SAT whose average-case running time is the one used in the paper. We have made this more clear in the revised paper.

Anyway, while the paper *cannot* be accepted in its current form, I actually think it’s interesting and could be suitable for Nature Communications, if (1) the above points are satisfactorily addressed and (2) the more experimentally-inclined reviewers agree.

We thank again the Reviewers for their insightful comments. We sincerely hope that our answers have fully addressed them.

Reviewers' Comments:

Reviewer #1:

Remarks to the Author:

I thank the authors for carefully taking into account my previous comments. In the new version of the manuscript, the authors have performed substantial revisions, and in particular they have clarified the context of their experiment that corresponds to solving an NP problem by having access only to limited information. Furthermore, the details of their protocol are now described in a clearer way.

As a last remark, given also the comments by the other reviewers concerning the context of their experimental demonstration, I am wondering whether it would be appropriate to consider a change in the title. More specifically, I would consider to explicitly cite in the title that the verification is performed in the scenario where only limited information is available. Thus, a possible suggestion for the title would be: "Experimental demonstration of quantum advantage for NP verification with limited information".

Even if this aspect is now clearly stated in the manuscript, this would avoid potential misleading interpretations of their results. Furthermore, this would be helpful to identify the task solved at a glance, given that by definition verification of NP problems with full information is a classically easy task. I would thus ask the authors to consider such title change (or analogous modification).

Besides this last comment, I am willing to support publication of this manuscript in Nature Communications.

Reviewer #2:

Remarks to the Author:

I would like to thank the authors for their thoughtful consideration of my comments on the previous manuscript. The re-working of the text, I believe, has made a substantial improvement to the manuscript that would ensure those reading it will not unintentionally misinterpret what is presented. The rewritten abstract is particularly good.

I think I am happy with how the authors have now presented the exact nature of the transition to quantum resources and the restriction to separable states.

The only reservation I still have of this work is on the overall significance. However, I would lean on the side of accepting this work as it has been a somewhat simple yet thought provoking work.

I have briefly looked over the other reviews and the author's addressing of the comments. I believe they have done a good job covering what are some technical points, yet keeping the manuscript at a broad readership level.

Reviewer #3:

Remarks to the Author:

I thank the authors for their revision, which addressed many of the concerns of me and the other reviewers. I had just a few more revisions:

In the abstract:

"while we also provide strong evidence that a classical computer would take much longer time (assuming only that it takes exponential time to solve an NP-complete problem)"

Add a clause like ", and assuming we fix the size of the witness." This is crucial to understanding everything that follows!

In the intro:

(we emphasize here that the quantum machine in our scenario is certainly not solving NP-complete problems but merely verifies them with limited information)

This is excellent, but "verifies them"  "verifies their solutions"?

In both their response and the revised manuscript, the authors claim that the best known classical running time for 3SAT "in the worst case" is 2^n . This is flatly incorrect: it's $\sim 1.3^n$. The $\sim 1.3^n$ algorithms of Schoning and others are not merely heuristic; rather, they're proven to have that performance on worst-case instances. Please revise accordingly!

Once these revisions are made, I recommend the paper for publication.

Nature Communications - Reply to Reviewer comments on “Experimental demonstration of quantum advantage for NP-verification with limited information”

January 6, 2021

2nd Round

1 Reviewer #1

I thank the authors for carefully taking into account my previous comments. In the new version of the manuscript, the authors have performed substantial revisions, and in particular they have clarified the context of their experiment that corresponds to solving an NP problem by having access only to limited information. Furthermore, the details of their protocol are now described in a clearer way.

As a last remark, given also the comments by the other reviewers concerning the context of their experimental demonstration, I am wondering whether it would be appropriate to consider a change in the title. More specifically, I would consider to explicitly cite in the title that the verification is performed in the scenario where only limited information is available. Thus, a possible suggestion for the title would be: “Experimental demonstration of quantum advantage for NP verification with limited information”. Even if this aspect is now clearly stated in the manuscript, this would avoid potential misleading interpretations of their results. Furthermore, this would be helpful to identify the task solved at a glance, given that by definition verification of NP problems with full information is a classically easy task. I would thus ask the authors to consider such title change (or analogous modification).

Besides this last comment, I am willing to support publication of this manuscript in Nature Communications. We thank the reviewer for their comments and positive recommendation. We agree with the suggestion and have accordingly changed the title of our manuscript to “Experimental demonstration of quantum advantage for NP-verification with limited information”.

2 Reviewer #2

I would like to thank the authors for their thoughtful consideration of my comments on the previous manuscript. The re-working of the text, I believe, has made a substantial improvement to the manuscript that would ensure those reading it will not unintentionally misinterpret what is presented. The rewritten abstract is particularly good.

I think I am happy with how the authors have now presented the exact nature of the transition to quantum resources and the restriction to separable states.

The only reservation I still have of this work is on the overall significance. However, I would lean on the side of accepting this work as it has been a somewhat simple yet thought provoking work.

I have briefly looked over the other reviews and the author’s addressing of the comments. I believe they have done a good job covering what are some technical points, yet keeping the manuscript at a broad readership level.

We thank the reviewer for their positive recommendation and for their previous comments that helped us improve our manuscript.

3 Reviewer #3

I thank the authors for their revision, which addressed many of the concerns of me and the other reviewers.

I had just a few more revisions:

In the abstract: “while we also provide strong evidence that a classical computer would take much longer time (assuming only that it takes exponential time to solve an NP-complete problem)” Add a clause like “, and assuming we fix the size of the witness.” This is crucial to understanding everything that follows!

In the abstract of the revised manuscript we have now written: “while we also provide strong evidence that, fixing the size of the proof, a classical computer would take much longer time”.

In the intro: (we emphasize here that the quantum machine in our scenario is certainly not solving NP-complete problems but merely verifies them with limited information) This is excellent, but “verifies them” \rightarrow “verifies their solutions”?

In the introduction we have now written: “we emphasize here that the quantum machine in our scenario is certainly not solving NP-complete problems but merely verifies whether a solution exists or not with limited information about the possible solution”.

In both their response and the revised manuscript, the authors claim that the best known classical running time for 3SAT “in the worst case” is 2^n . This is flatly incorrect: it’s $\sim 1.3^n$. The $\sim 1.3^n$ algorithms of Schoning and others are not merely heuristic; rather, they’re proven to have that performance on worst-case instances. Please revise accordingly!

Under the subheading ‘Classical complexity of verification’ we have now written: “In particular, we consider the classical complexity to be of the form $2^{\gamma N}$ for some constant $\gamma \leq 1$. Shöning’s algorithm for 3-SAT takes time $(4/3)^n$ on average for instances of size n [1], while the best-known practical SAT solvers can provide a complexity of $O(1.307^N) = O(2^{0.4N})$ [2].” We have added the reference proposed by the reviewer, but the one we used is actually the fastest SAT solver known in literature.

Once these revisions are made, I recommend the paper for publication.

We thank the reviewer for their positive recommendation and for their comments in both reviewing rounds that helped us improve our manuscript.

References

- [1] U. Schöning, IEEE Symposium of Foundations of Computer Science (FOCS) (1999).
- [2] T. D. Hansen, H. Kaplan, O. Zamir, and U. Zwick, in Proceedings of the 51st Annual ACM SIGACT Symposium on Theory of Computing (2019), pp. 578–589.